# Tumor-Associated Neutrophils Are a Negative Prognostic Factor in Early Luminal Breast Cancers Lacking Immunosuppressive Macrophage Recruitment

**DOI:** 10.3390/cancers16183160

**Published:** 2024-09-15

**Authors:** Eva Schmidt, Luitpold Distel, Ramona Erber, Maike Büttner-Herold, Marie-Charlotte Rosahl, Oliver J. Ott, Vratislav Strnad, Carolin C. Hack, Arndt Hartmann, Markus Hecht, Rainer Fietkau, Sören Schnellhardt

**Affiliations:** 1Department of Radiation Oncology, Universitätsklinikum Erlangen, Friedrich-Alexander-Universität Erlangen-Nürnberg, 91054 Erlangen, Germany; evaschmidt1996@web.de (E.S.);; 2Comprehensive Cancer Center Erlangen-EMN (CCC ER-EMN), 91054 Erlangen, Germany; 3Institute of Pathology, Universitätsklinikum Erlangen, Comprehensive Cancer Center Erlangen-EMN, Friedrich-Alexander-Universität Erlangen-Nürnberg, 91054 Erlangen, Germany; 4Department of Nephropathology, Institute of Pathology, Universitätsklinikum Erlangen, Friedrich-Alexander-Universität Erlangen-Nürnberg, 91054 Erlangen, Germany; 5Department of Gynecology and Obstetrics, Universitätsklinikum Erlangen, Comprehensive Cancer Center Erlangen-EMN, Friedrich-Alexander-Universität Erlangen-Nürnberg, 91054 Erlangen, Germany; 6Department of Radiotherapy and Radiation Oncology, Saarland University Medical Center, 66421 Homburg, Germanysoeren.schnellhardt@uks.eu (S.S.)

**Keywords:** breast cancer, luminal, tumor-associated neutrophils, CD66b, tumor-associated macrophages

## Abstract

**Simple Summary:**

Neutrophil granulocytes in the vicinity of malignant tumors are referred to as tumor-associated neutrophils (TANs). Knowledge about the role of TANs in the disease progression of early hormone-receptor-positive breast cancer is limited but essential to the development of new immunotherapies and biomarkers. In this work, we counted immunohistochemically stained TANs in sections of 144 early-stage breast cancer tumors and correlated these results with disease-free survival. Our results indicate that not only intratumoral TANs but also those in adjacent normal tissue and in sentinel lymph nodes were associated with shorter disease-free survival. Combined analysis with other immune cells from previous studies revealed that intratumoral TANs were only associated with prognosis in tumors that did not express an unfavorable macrophage polarization profile, providing a clinical example of the known interactions between different types of immune cells within the tumor microenvironment. This indicates that future research in the field should evaluate the two types of immune cells together.

**Abstract:**

Background: Tumor-associated neutrophils (TANs) are important modulators of the tumor microenvironment with opposing functions that can promote and inhibit tumor progression. The prognostic role of TANs in early luminal breast cancer is unclear. Methods: A total of 144 patients were treated for early-stage hormone-receptor-positive breast cancer as part of an Accelerated Partial Breast Irradiation (APBI) phase II trial. Resection samples from multiple locations were processed into tissue microarrays and sections thereof immunohistochemically stained for CD66b+ neutrophils. CD66b+ neutrophil density was measured separately in the stromal and intraepithelial compartment. Results: High stromal and intraepithelial CD66b+ TAN density was a negative prognostic factor in central tumor samples. In addition, neutrophil density in adjacent normal breast tissue and lymph node samples also correlated with reduced disease-free survival. TAN density correlated with CD163+ M2-like tumor-associated macrophage (TAM) density, which we analyzed in a previous study. TANs were a negative prognostic factor in tumors with an elevated M1/M2 TAM ratio, while this impact on patient outcome was lost in tumors with a low M1/M2 ratio. A combined multivariate analysis of TAM and TAN density revealed that only TAM polarization status was an independent prognostic factor. Conclusions: CD66b+ neutrophils were a negative prognostic factor in early-stage luminal breast cancer in single-marker analysis. Combined analysis with TAMs could be necessary to correctly evaluate their prognostic impact in future studies. TAN recruitment might act as a compensatory mechanism of immunoevasion and disease progression in tumors that are unable to sufficiently attract and polarize TAMs.

## 1. Introduction

With an estimated lifetime risk of over 10%, breast cancer remains the most common type of malignancy in women. Due to medical advances and extensive screening programs, many cases are detected early and can be treated successfully [1]. For this reason, a large focus of clinical research regarding the prognostically favorable early luminal molecular subtype has been on therapy de-escalation. Techniques such as partial breast irradiation allow a reduction in therapy-related toxicities while maintaining a very high level of disease control [2]. These de-escalated treatment strategies do, however, require careful patient selection to avoid undertreatment of tumors with a more aggressive biology.

A key aspect of the tumor microenvironment that could improve our understanding of tumor biology and simultaneously act as potent biomarker for treatment planning is tumor-infiltrating lymphocytes (TILs) and other tumor-infiltrating inflammatory cells (TIICs). TIICs can have a crucial role both in disease progression and in tumor control and in this regard, not only the type of immune cell is of importance, but also its localization and distribution patterns within the tumor [3,4]. Within the context of different immune phenotypes, diverse mechanisms of intratumoral immunosuppression come into play and one type of immune cell may exert completely opposite functions [5].

Tumor-associated neutrophil granulocytes (TANs) are an important example of the differing roles of a single type of inflammatory cell. Traditionally, TANs were considered an anti-tumor component of the tumor microenvironment due to their cytotoxic abilities like H_2_O_2_ secretion. However, in recent years, numerous progression-promoting mechanisms have been described: TANs contribute to tumorigenesis and tumor growth, promote angiogenesis, engage in immunosuppression and inhibit cytotoxic T cells, contribute to the formation of the premetastatic niche, and use neutrophil extracellular traps (NETs) to direct circulating tumor cells to distant organ sites [6].

Due to these opposing functions, a classification into two distinct polarization states similar to those of tumor-associated macrophages (TAMs) has been established. Under the influence of TGF-beta and G-CSF, the tumor-promoting N2 phenotype is expressed, whereas IFN-beta promotes the anti-tumor N1 phenotype [7]. However, in contrast to TAMs, no reliable single markers exist for these two phenotypes, preventing a direct immunohistochemical detection of the polarization status of a TAN population.

Consequently, it is not surprising that increased infiltration with TANs has been associated with both improved and reduced prognosis in various solid tumors. Overall, however, studies have found TANs to predominate as a negative prognostic factor in most cancers and TANs are frequently described as a potential biomarker [8].

Data on the prognostic role of TANs in breast cancer are mixed and further clarification is needed: Boissiere-Michot et al. reported no prognostic relevance of CD66b+ TANs in a mixed subtype as well as a triple-negative breast cancer (TNBC) cohort [9,10]. Geng et al. showed that in tumors treated with neoadjuvant chemotherapy, an increase in CD66b+ cells was prognostically unfavorable, but there were no conclusive results regarding pretherapeutic TAN density [11]. Wang et al. showed a negative prognostic effect, especially in TNBC, of parenchymal CD66b+ TANs only [12]. Two other studies also found a correlation between CD66b+ TAN density and worsened prognosis in cohorts with mixed subtypes [12,13]. A consistent finding of the above-mentioned studies was that the various molecular subtypes of breast cancer displayed different degrees of TAN infiltration, and that infiltration was most pronounced in TNBC. This highlights that an evaluation of prognostic relevance as well as the establishment of cut-off values for prognostic groups should be performed separately according to subtypes. Overall, the triple-negative subtype is more often the focus of immunological research, since it is considered more immunogenic and has a higher rate of treatment failure and thus a more urgent need for new therapies [14,15]. The luminal subtype on the other hand, despite being the most common subtype of breast cancer, is less frequently investigated due to its better prognosis and the poorly understood impact of immunological factors on disease outcome [16]. A large meta-analysis on the prognostic impact of TILs in general demonstrated the need for further immunological studies in the luminal subtype: while TILs were clearly associated with improved outcome in TNBC and the Her2+/HR- subtype, the opposite was the case in luminal breast cancer and TILs were a negative prognostic factor, implying fundamental differences in TIL behavior in this subtype [17]. Possible explanations range from the impact of antihormonal therapies on TIICs to a reduced expression of neoantigens to TIICs being a passive bystander of a more aggressive subgroup of luminal breast cancers [16,18].

The subtype dependence of the prognostic relevance of TILs and TANs and the fact that patient cohorts in previous studies on TANs were mixed in subtype or triple-negative was a key reason for conducting the present analysis, which investigated the prognostic impact of TANs on the luminal subtype only. The group of patients whose tumor tissue was studied here participated in a phase II trial on accelerated partial breast irradiation (APBI). The strict trial inclusion criteria guaranteed a cohort of patients with exclusively low-risk early luminal breast cancer and high-quality, long-term prospective clinical data on disease outcome. This very uniform group of patients with minimal confounding factors provided optimal conditions to clarify the role of TANs in early luminal breast cancer. In previous works, we already demonstrated the surprisingly clear prognostic relevance of M1-like and M2-like TAMs as well CD4+ T cells, CD20+ B cells, and CD45RO+ memory T cells in this same cohort of patients [19,20]. Thus, apart from evaluating the prognostic relevance of immunohistochemically stained CD66b+ TANs, this analysis also allowed us to connect these results to those of other TIICs from our previous studies to draw possible conclusions about interactions with other immune cells and the mechanisms of immunosuppression.

## 2. Materials and Methods

### 2.1. Patients and Clinical Data

A total of 144 patients were treated for early stage hormone-receptor-positive breast cancer at Universitätsklinikum Erlangen as part of the German–Austrian Accelerated Partial Breast Irradiation (APBI) phase II trial [2]. As previously reported, the main trial inclusion criteria were histopathologically confirmed invasive breast carcinoma of any histologic subtype of ≤3 cm in diameter, clear resection margins of ≥ 2 mm in any direction, hormone sensitivity (estrogen receptor-positive (ER+)/progesterone receptor-positive (PR+), ER+/PR−, ER−/PR+), histologic grade 1 or 2, no lymph vessel and no blood vessel invasion, no or only microscopically involved axillary nodes (pN0/pNmi), no distant metastases, and age ≥35 years [2]. Interstitial multicatheter pulsed dose rate (PDR) or high dose rate (HDR) brachytherapy were performed in all patients after breast conserving surgery. Resection samples were used for tissue microarray (TMA) construction. Most patients (90.3%) received adjuvant hormone therapy, ten (6.9%) were treated with chemotherapy, and eight patients received both treatments (Table 1).

### 2.2. TMA Construction and Immunohistochemistry

Tissue microarrays with a diameter of 2 mm per core were processed from each of the 144 paraffin-embedded tumor resections. Since it is unknown to which distance the influence of the tumor microenvironment on TANs extends, samples from the tumor resections were taken from different locations, beginning with the central tumor (CT) and then moving outwards from the invasive front of the tumor (IF), normal tissue in tumor proximity (prox), normal tissue distant from the tumor (dist), and finally also from the resected lymph nodes. Tissue sections (2 µm) were de-paraffinized in xylene and rehydrated with graded ethanol. CD66b immunohistochemical stainings were performed using a mouse monoclonal anti-human CD66b antibody (clone G10F5, dilution 1:200) (555723, BD Biosciences, Heidelberg, Germany) and a goat anti-mouse IgM secondary antibody (Vector Laboratories, Newark, CA, USA), as described previously [21].

Stained TMAs were digitalized on a high-throughput scanner (Mirax Scan, Zeiss, Göttingen, Germany) and processed digitally in Pannoramic Viewer (3D Histech, Budapest, Hungary). While samples were available from 144 patients in total, some biopsies only contained stromal or epithelial tissue, resulting in fewer available datasets than the total number of patients. In particular, the staining process of normal tissue biopsies which often contained a high degree of fat led to a very low number of evaluable samples.

The definitions released by the St Gallen International Expert Consensus on the Primary Therapy of Early Breast Cancer were used to classify intrinsic breast cancer subtypes [22]. Ki67 expression levels of ≥20% were described as an unfavorable prognostic factor in early breast cancer by Fasching et al. [23].

### 2.3. Quantification of Neutrophils

As described previously, CD66b cell densities were counted semi-automatically with image processing software (Biomas software, version 3.3, Erlangen, Germany). Inclusion criteria were size, morphology, and color. The respective areas of the stromal and intraepithelial compartments were registered and cell densities were analyzed for each compartment separately to account for possible differences in TAN function in the two compartments.

### 2.4. Cell Densities of Other Tumor-Infiltrating Inflammatory Cells

Data on the cell densities of M1-like (CD68+/CD163−) and M2-like (CD68+/CD163+) TAMs, CD4+ T helper cells, CD45RO+ memory T cells, CD1a+ dendritic cells, and CD20+ B cells in this cohort of patients have been published previously [19,20].

### 2.5. Statistical Analyses

Statistical analyses were performed in SPSS version 27 (IBM Inc., Chicago, IL, USA). Correlations were identified through Spearman’s Rho and the Chi-squared test. Mean values of cell densities were compared with Student’s *t*-test for independent samples and Welch’s test. The Cox proportional hazards model was used to calculate hazard ratios. Covariates with *p* < 0.15 in univariate analysis were included in multivariate analyses. The proportional hazards assumption was verified by visual examination of the log-minus-log curves. Optimal cut-off points for prognostic groups based on CD66b+ cell density were calculated for disease-free survival (DFS) through receiver operating characteristic (ROC) curve analysis and confirmed via X-tile software (version 3.6.1, Yale School of Medicine, New Haven, CT, USA) [24]. The Kaplan–Meier method was used to plot survival curves. Estimated survival times were compared with the log-rank test. *p*-values < 0.05 were considered to be statistically significant.

## 3. Results

The studied cohort consisted of 144 patients with early-stage breast cancer and mainly T1, N0/UICC I stage (Table 1) [2].

### 3.1. CD66b+ Cell Density in Different Locations

Immunohistochemically stained CD66b+ neutrophils and total epithelial and stromal areas were detected in biopsies from the central tumor and invasion front (Figure 1A), resected lymph nodes, (Figure 1B), and adjacent normal tissue in the immediate vicinity of the tumor (prox) or at the margin of the resected tissue (dist) (Figure 1C).

A high variance in measured cell densities was observed in all localizations, with a particularly large number of outliers concerning stromal CD66b+ cell densities in tumor samples (Figure 1D). In both the central tumor (median 4.0 cells/mm^2^, SD: 185.2 cells/mm^2^) and the invasion front (median 4.3 cells/mm^2^, SD: 265.4 cells/mm^2^), significantly higher CD66b+ cell densities were measured in the stroma compared to the tumor epithelium (*p* < 0.05). The highest median cell density was measured in the lymph nodes with 25.1 cells/mm^2^ (SD: 53.1 cells/mm^2^).

### 3.2. CD66b+ Neutrophils as a Prognostic Factor

For the subgroup of patients of the German–Austrian APBI trial that was studied here, disease-free survival (DFS) at 10 years was 87.2% (15-yr DFS: 81.4%) (Figure 2A).

Cut-off points for prognostic groups based on CD66b+ cell density were calculated through receiver operating characteristic (ROC) curve analysis. Elevated TAN density (≥74.8 cells/mm^2^) in the lymph nodes identified a small group of patients (*n* = 9) with a strongly increased risk of disease recurrence or death (Figure 2B). In normal tissue samples from tumor proximity (n = 30), a high stromal CD66b+ cell density was a negative prognostic factor (*p* = 0.037) (Figure 2C), while in normal tissue from the periphery of resected tissue, only a weak trend towards worse prognosis was observed (*p* = 0.250) (Figure 2D).

In central tumor samples, both high stromal (*p* < 0.001) and intraepithelial (*p* = 0.025) densities of CD66b+ cells were associated with significantly reduced disease-free survival (Figure 3A,B). In samples from the invasive front, only a trend to this effect could be detected (stromal: *p* = 0.215; intraepithelial: *p* = 0.155) (Figure 3C,D).

### 3.3. Correlations with Clinical Characteristics

Patients were grouped according to high and low stromal and intraepithelial CD66b+ cell density in central tumor and lymph node samples with cut-off values as defined by the prognostic groups in Figure 2 and Figure 3. These groups were then correlated with clinical characteristics. Stromal CD66b+ cell density in central tumor samples only had an inverse correlation with the presence of DCIS; otherwise, there were no correlations with clinical parameters for both stromal and intraepithelial central tumor CD66b+ cell densities. A high CD66b+ cell density in the lymph nodes, on the other hand, correlated significantly with luminal B subtype, a high proliferation index, and lobular histology (Table 2). A corresponding correlation analysis was performed with groups based on high and low CD66b+ cell density in central tumor samples and samples from outside the tumor area. CD66b+ cell density in lymph node and normal tissue samples did not correlate with CD66b+ cell density in central tumor samples.

### 3.4. Correlations with Other Tumor-Infiltrating Inflammatory Cells

In two previous independent immunohistochemical studies, we investigated the prognostic impact of a range of other TIICs in this same cohort with a similar methodology [19,20]. These inflammatory cells included M1-like (CD68+/CD163−) and M2-like (CD68+/CD163+) TAMs, CD4+ T helper cells, CD45RO+ memory T cells, CD1a+ dendritic cells, and CD20+ B cells. Densities of all cell types except for dendritic cells were associated with DFS, with TAMs having the most consistent relationship with prognosis. TANs have so far not been analyzed in this cohort. Our current analysis allowed us to correlate CD66b+ TAN densities with those of other TIICs measured in previous studies in central tumor and invasive front samples in this group of patients.

Intraepithelial and stromal CD66b+ cell densities in the central tumor and invasion front correlated most consistently with M2-like macrophage densities. A correlation of neutrophils with CD4+ T helper cells, CD45RO+ memory T cells, and CD20+ B cells was observed in the stromal compartment of central tumor samples (Table 3).

### 3.5. Prognostic Impact According to Macrophage Polarization Status

We divided the cohort according to stromal and intraepithelial macrophage polarization status, which we defined in our previous work, and thus established two groups: a prognostically favorable group with a high M1/M2 ratio (stromal and intraepithelial median M1/M2 ratio: 0.061 and 0.103, respectively) and an overall prognostically very unfavorable group with a low M1/M2 ratio (stromal and intraepithelial median M1/M2 ratio: 0.00 and 0.007, respectively) [19]. Average stromal and intraepithelial CD66b+ cell density in central tumor samples (Figure 4A) was higher in the M1/M2 low group, but there was no statistically significant difference in cell density distribution.

In central tumor samples with a high ratio of M1/M2 macrophages, TANs remained a negative prognostic factor (stromal *p* < 0.001, intraepithelial *p* = 0.032) (Figure 4B,C), while in tumors with a low M1/M2 ratio no prognostic relevance of stromal and intraepithelial CD66b+ cell densities was observed (Figure 4D,E). Stromal and intraepithelial CD66b+ cell densities in invasive front samples remained without clear prognostic significance after subdivision into groups based on M1/M2 ratio (Appendix A).

### 3.6. Multivariate Cox Regression Analysis

In the single-marker multivariate Cox regression analysis, stromal CD66b+ TAN density in central tumor (*p* = 0.027) and lymph node samples (*p* = 0.002) were independent prognostic factors (Table 4). In the multivariate Cox regression analysis that included macrophage polarization status, this association was lost and only intraepithelial macrophage polarization status (*p* = 0.038) was an independent prognostic factor (Table 5). Cohort size was not sufficient for Cox regression of subgroups according to M1/M2 ratio.

## 4. Discussion

The disease-modulating properties of tumor-infiltrating lymphocytes (TILs) and other inflammatory cells (TIICs) continue to be of great research interest in breast cancer, both on a mechanistic level and as prognostic biomarkers [25,26]. Although triple-negative breast cancer (TNBC) is more frequently addressed in this context, we have already been able to demonstrate the surprisingly strong prognostic relevance of various TIICs in the cohort of patients with early luminal breast cancer investigated here [14,16,19,20]. In the present study, we extended this analysis with the addition of CD66b+ tumor-associated neutrophil granulocytes (TANs).

### 4.1. TAN Function in the Tumor Microenvironment

Under the influence of cytokines like TGF-β, G-CSF, and interferon-β, TANs, similar to TAMs, can exhibit different polarization states and thus exert both anti-tumor (N1) and pro-tumor (N2) effects in the tumor microenvironment [7]. N1-polarized TANs can destroy tumor cells through the secretion of reactive oxygen species and have been reported to participate in T cell activation. Simultaneously, N2-TANs initiate tumor cell growth as well as angiogenesis via a CD90-TIMP-1 loop and the G-CSF-RLN2-MMP-9 axis in breast cancer and suppress a cytotoxic immune response [6,12,13]. Moreover, in breast cancer estrogen alters the activity and gene expression of neutrophils to contribute to tumor formation and growth [27,28]. Furthermore, TANs might also play a role in resistance to radiotherapy, which is a key element of breast-conserving treatment strategies in breast cancer [29].

Neutrophil extracellular traps (NETs), a complex of decondensed DNA and various proteins that physiologically contributes to the containment and elimination of microbial threats, have recently attracted considerable attention as another important instrument of tumor progression used by pro-tumor TANs. NETs can not only shield tumor cells from interaction with cytotoxic immune cells, but also induce tumor cell reactivation and proliferation and promote the spread and transition of distant metastasis through induction of thrombosis and epithelial-to-mesenchymal transition [30].

Studies suggest that the role and polarization of TANs might be related to tumor stage, with early cancers being infiltrated by N1 anti-tumor TANs and a shift towards N2 recruitment in more advanced disease stages. Most of the in vivo evidence for this thesis stems from studies in colorectal cancer, where TANs were a positive prognostic factor in early-stage disease. This hypothesis, however, is not applicable to all types of cancer, as in early-stage melanoma and cervical cancer intratumoral neutrophils were associated with poor prognosis [7].

### 4.2. Prognostic Impact of TANs in Early Luminal Breast Cancer

The results of our work suggest that increased CD66b+ cell density both in the stromal and intraepithelial compartment of central tumor tissue from early luminal breast cancers is a negative prognostic factor associated with tumor progression. This is consistent with the above-mentioned results of studies in other subtypes of breast cancer [9,10,12,13]. In addition, we also made intriguing novel observations in tissue beyond the primary tumor: Although there was limited availability of normal tissue samples in this study, a negative prognostic relevance of CD66b+ cells was also observed in these samples. And even in surgically removed lymph nodes, which were free from metastasis, we could identify a small group of patients with a strongly increased CD66b+ cell density and significantly elevated long-term risk for recurrence or distant metastasis. Overall, central tumor samples appeared to be the location with the most relevant association between CD66b+ neutrophil density and reduced prognosis. Due to the limited cohort size, further statistical analysis comparing the prognostic relevance of different locations was not performed.

Correlation of cell densities with clinical parameters also revealed that neutrophil infiltration in the lymph nodes correlated with more aggressive clinical parameters. Thus, in the luminal subtype of breast cancer, increased infiltration with CD66b+ neutrophils in the primary tumor itself, as well as in adjacent normal tissue and lymph nodes, was associated with worse disease outcome. This suggests that the ability of breast cancers to recruit neutrophil granulocytes and to influence their behavior may extend far beyond the classical tumor microenvironment. Our results also suggest that the hypothesis of an anti-tumor role of TANs in early-stage disease seems to be limited to certain tumor entities and does not apply to luminal breast cancer, as we observed a clear negative prognostic impact in a cohort of patients suffering exclusively from early-stage luminal breast cancer.

### 4.3. Combined Analysis of TANs and TAMs

The mechanisms by which TANs contribute to cancer progression have been a subject of ongoing oncoimmunological research in recent years. We were particularly interested in the aspect of immunosuppression and possible interactions with other inflammatory cells, which we already quantified in previous studies. We found the most consistent intratumoral correlation between M2-like TAM density and CD66b+ TAN density. This is in contrast to the results of Boissiere-Michot et al., who could not report any association between the density of TAMs and TANs in their work [9]. However, in their study, the pan-macrophage marker CD68 was applied, whereas here CD163 was used to specifically identify immunosuppressive M2-like TAMs. Thus, there may be a dependency or mutual recruitment between M2-like TAMs and TANs, a process which has also been described in the literature [31]. In hepatocellular cancer, Zhou et al. reported the recruitment of tumor-associated macrophages by TANs as a mechanism of tumor progression [32]. Tumor-induced G-CSF release, which as described by Sheng et al. promotes metastasis in TANs via the G-CSF-RLN2-MMP-9 axis, also affects the polarization of TAMs to the immunosuppressive M2-like phenotype [13].

Further information on the possible interaction between TANs and TAMs in our study can be derived from multivariate survival analysis. In the single-marker multivariate Cox regression analysis, CD66b+ TAN density was an independent prognostic factor, but this prognostic relevance was lost when TAM polarization status was included, indicating that TAMs might be the dominant factor determining disease outcome in early luminal breast cancer. Moreover, this emphasizes another important point: while single-marker analysis of TIICs as potential prognostic biomarkers is commonly performed in the literature due to its feasibility and ease of use in the clinical setting, results and their functional implications should always be interpreted with caution due to the complex interactions of different types of TIICs in the tumor microenvironment.

What could a possible complementary pro-tumor dynamic between TAMs and TANs look like? One of the main observations of our previous work on TAMs was that in early luminal breast cancer, a combined analysis of cytotoxic M1-like and immunosuppressive M2-like TAMs, i.e., the overall polarization status of the macrophage population, had the most pronounced association with prognosis [19]. A particularly unfavorable constellation of macrophage polarization was the combination of high M2-like density and low M1-like density (M2-shifted) or, in other words, a low M1-like/M2-like ratio. This was contrasted by patients with tumors with a higher M1/M2 ratio and almost ideal response to therapy. Interestingly, dividing the cohort into two groups based on the M1/M2 ratio in our present study revealed that within the group with a low M1/M2 ratio, CD66b+ TAN density was no longer prognostically relevant. In the remaining tumors with higher M1/M2 ratios, CD66b+ TANs were still significantly associated with reduced disease-free survival. This may reflect alternative primary mechanisms of immunosuppression: while some tumors succeed in recruiting and repolarizing the macrophage population to the suppressive M2-like phenotype (low M1/M2 ratio, Type A), others primarily utilize the progression-promoting properties of TANs (Type B) (Figure 5). These tumors express a prognostically favorable M1/M2 ratio but the anti-tumor effects of the M1-like TAM phenotype are counteracted by suppressive TANs.

In experimental studies in both cervical carcinoma and pancreatic adenocarcinoma, results suggest that there is a complementary immunosuppressive dynamic between the two myeloid cell types [33,34]. Macrophage depletion via CCR2 blockade led to compensatory influx of neutrophils into the tumor, and only blockade of both cell types led to initiation of a tumor-directed immune response. A similar mechanism may also be relevant in early luminal breast cancer. Due to the immunohistochemical methodology of the present study, this hypothesis is of course speculative at this point and requires experimental confirmation.

### 4.4. Strengths and Limitations

The strengths of this study were the precise detection of cell densities separately in the stromal and intraepithelial compartments, as well as the uniform clinical characteristics of the patient cohort with exclusively luminal breast carcinoma, which were treated within the framework of a clinical trial. The relatively small cohort of patients with a low number of recurrences and metastases entailing a degree of statistical uncertainty was the major weakness of our analysis. Another limitation was the fact that CD66b immunohistochemistry does not allow a distinction between anti-tumor N1- and pro-tumor N2-TANs.

## 5. Conclusions

In the single-marker analysis, increased density of CD66b+ TANs in the central tumor, but also in adjacent normal tissue and the lymph nodes, was predictive of reduced disease-free survival in early luminal breast cancer. TAN density in the lymph nodes was associated with more aggressive disease characteristics.

Intratumoral neutrophil density correlated with M2-like macrophage density. Combined multivariate survival analysis revealed that TAM polarization status alone was an independent prognostic factor. Subgroup analysis indicated that neutrophil infiltration was only associated with reduced prognosis in tumors with an increased M1/M2 TAM ratio. TAN recruitment might act as a compensatory mechanism of immunoevasion and disease progression in tumors which are unable to sufficiently attract and polarize TAMs. Further immunohistochemical and experimental studies are required to understand the combined impact of intratumoral neutrophils and macrophages on disease outcome in early luminal breast cancer.

## Figures and Tables

**Figure 1 cancers-16-03160-f001:**
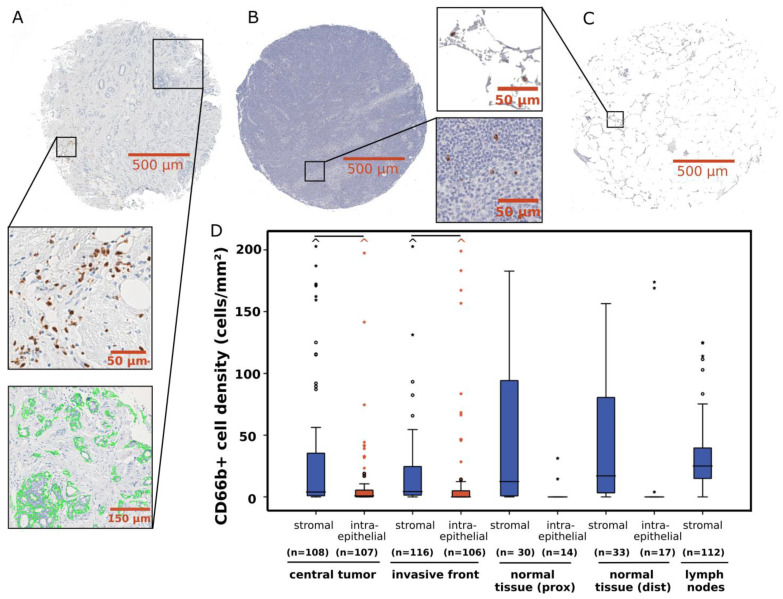
Representative images of sections of breast cancer tissue microarray cores with immunohistochemical CD66b staining. (**A**) Invasive front sample with stromal neutrophil infiltration (brown). Example of intraepithelial compartment segmentation (green) via BIOMAS software. (**B**) Tissue microarray core of lymph node sample with an example of stained neutrophils (brown) among other inflammatory cells (blue). (**C**) Normal tissue sample with immunohistochemical CD66b staining of neutrophils (brown) in between adipose tissue. (**D**) Box plots of stromal and intraepithelial CD66b+ cell density distribution in different locations. Horizontal black bars signify *p* < 0.05 in Student’s *t*-test. The central line indicates median values while the box represents the interquartile range (IQR). Whiskers represent 1.5 × IQR or minimum/maximum. Outliers are represented by dots (up to 3 × IQR) or asterisks (>3 IQR). Outliers not visible in this diagram are depicted as a caret.

**Figure 2 cancers-16-03160-f002:**
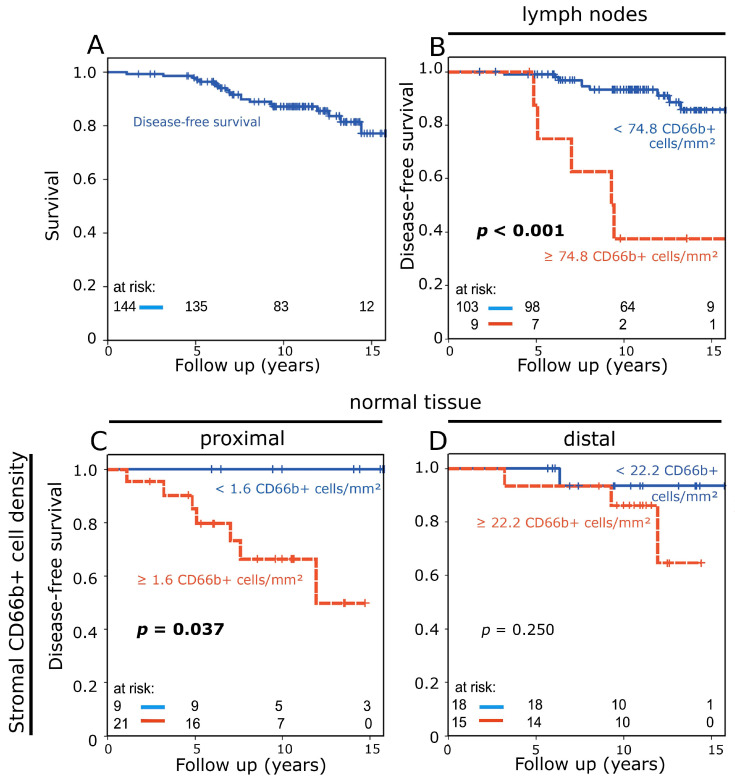
(**A**) Disease-free survival of the studied patient cohort analyzed with the Kaplan–Meier method and log-rank test. (**B**) Disease-free survival according to stromal CD66b+ neutrophil density in lymph nodes. (**C**,**D**) Disease-free survival according to stromal CD66b+ neutrophil density in normal tissue from tumor proximity (**C**) and normal tissue from the periphery of the resection (**D**).

**Figure 3 cancers-16-03160-f003:**
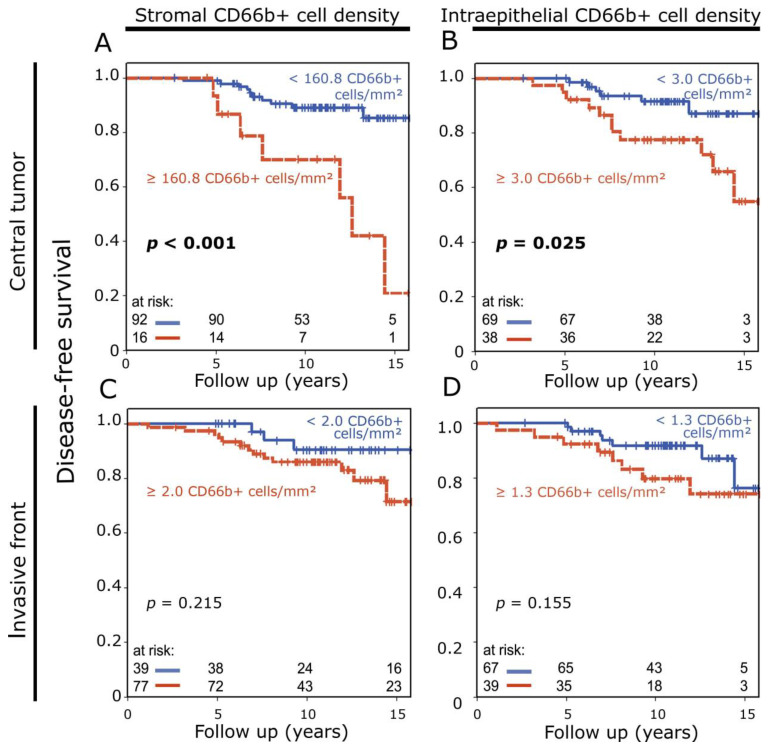
(**A**,**B**) Disease-free survival rate according to stromal (**A**) and intraepithelial (**B**) CD66b+ neutrophil density in central breast tumor samples analyzed with the Kaplan–Meier method and log-rank test. (**C**,**D**) Disease-free survival rate according to stromal (**C**) and intraepithelial (**D**) CD66b+ neutrophil density in invasive front samples.

**Figure 4 cancers-16-03160-f004:**
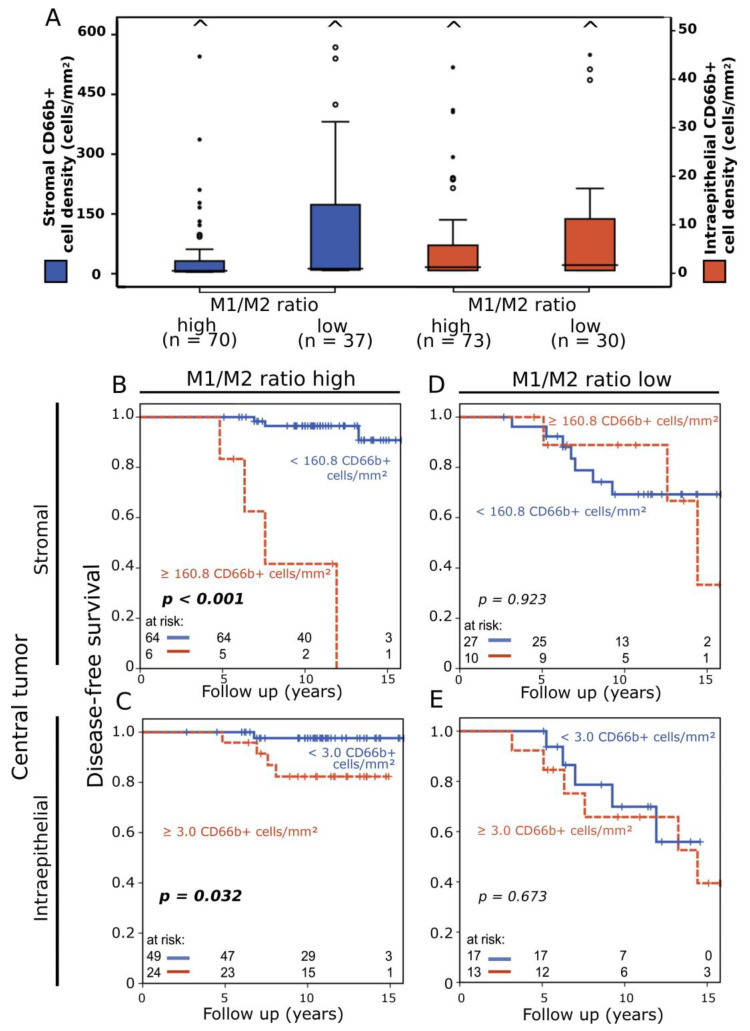
(**A**) Box plots of stromal (blue) and intraepithelial (red) CD66b+ cell density distribution in patients with high versus low M1/M2 ratio. Carets signify outliers. (**B**,**C**) Disease-free survival rate according to stromal (**B**) and intraepithelial (**C**) CD66b+ neutrophil density in central breast tumor samples in patients with a high M1/M2 ratio analyzed with the Kaplan–Meier method and log-rank test. (**D**,**E**) Disease-free survival according to stromal (**D**) and intraepithelial (**E**) CD66b+ neutrophil density in central tumor samples in patients with a low M1/M2 ratio.

**Figure 5 cancers-16-03160-f005:**
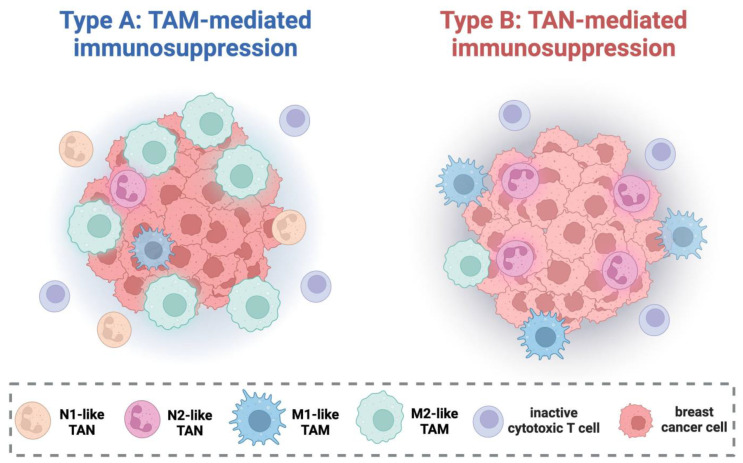
Proposed model of cellular mechanisms of immunosuppression in early luminal breast cancer. Type A tumors are capable of polarizing tumor-associated macrophages (TAMs) towards the immunosuppressive M2-like state (low M1/M2 ratio), resulting in inactivation of anti-tumor elements like cytotoxic T cells. Tumor-associated neutrophils (TANs) are numerous but play a subordinate role, possibly due to the contrary effects of anti-tumor N1-like TANs and suppressive N2-like TANs in this scenario. Type B tumors are not capable of macrophage repolarization (high M1/M2 ratio) and utilize the immunosuppressive properties of TANs which are most likely polarized towards the N2-like state. Figure created with Biorender.com.

**Table 1 cancers-16-03160-t001:** Clinical characteristics of the studied patient cohort.

Clinical Characteristics	Categories
Age (yr)	Mean: 59.0; <50: 28 (19.4%); ≥50: 116 (80.6%)
T category	pT1a: 9 (6.3%); pT1b: 37 (25.7%); pT1c: 82 (56.9%); pT1mic: 6 (4.2%); pT2: 10 (6.9%)
N category	N0: 141 (97.9%); N1: 3 (2.1%)
Stage	UICC I: 132 (91.7%); UICC II: 12 (8.3%)
Tumor size (mm)	<10: 38 (26.4%); 10–20: 96 (66.7%); >20: 10 (6.9%)
Histological grading	G1: 37 (25.7%); G2: 100 (69.4%); G3: 4 (2.8%); n.a. 3 (2.1%)
Histological typing	lobular: 22 (15.3%); no special type: 100 (69.4%); other: 22 (15.3%)
Ki67 (%)	<20: 109 (75.7%); ≥20: 31 (21.5%); n.a.: 4 (2.8%)
ER status	positive: 139 (96.5%); negative: 1 (0.7%); n.a.: 4 (2.8%)
PR status	positive: 130 (90.3%); negative: 11 (7.6%); n.a.: 3 (2.1%)
Her2 status	positive: 7 (4.9%); negative: 132 (91.7%); n.a.: 5 (3.5%)
Subtype	Luminal A: 96 (66.7%); Luminal B: 43 (29.9%); n.a.: 5 (3.5%)
Hormone therapy	Yes: 130 (90.3%); No: 14 (9.7%)
Chemotherapy	Yes: 10 (6.9%); No: 134 (93.1%)

**Table 2 cancers-16-03160-t002:** Association between clinicopathological characteristics and CD66b+ neutrophil density in both tumor compartments of central tumor samples and lymph nodes.

		Stromal (N = 108)			Intraepithelial (N = 107)			Lymph Node (N = 112)		
	N (Total)	CD66b+ TAN Density Low	CD66b+ TAN Density High	*p*	CD66b+ TAN Density Low	CD66b+ TAN Density High	*p*	CD66b+ TAN Density Low	CD66b+ TAN Density High	*p*
Age (yr)				0.52			0.48			0.37
<50	28	22 (24%)	2 (13%)		14 (20%)	10 (26%)		18 (17%)	3 (33%)	
≥50	116	70 (76%)	14 (87%)		55 (80%)	28 (74%)		85 (83%)	6 (67%)	
Stage				0.64			1.00			0.18
UICC I	132	84 (91%)	14 (88%)		63 (91%)	36 (95%)		95 (92%)	7 (78%)	
UICC II	12	8 (9%)	2 (12%)		6 (9%)	2 (5%)		8 (8%)	2 (22%)	
Tumor size (mm)				1.00			0.71			0.50
<20	134	85 (92%)	15 (94%)		62 (93%)	36 (92%)		96 (93%)	8 (89%)	
≥20	10	7 (8%)	1 (6%)		5 (7%)	3 (8%)		7 (7%)	1 (11%)	
Histological grading				1.00			0.07			0.11
G1	37	26 (29%)	4 (25%)		24 (35%)	6 (17%)		27 (26%)	0 (0%)	
G2 + G3	104	63 (71%)	12 (75%)		44 (65%)	30 (83%)		76 (74%)	9 (100%)	
n.a.	3									
Histological typing				0.67			0.54			0.04
non-lobular	122	83 (90%)	14 (88%)		65 (94%)	32 (84%)		88 (85%)	5 (56%)	
lobular	22	9 (10%)	2 (12%)		4 (6%)	6 (16%)		15 (15%)	4 (44%)	
DCIS				0.02			0.20			0.48
no	76	38 (46%)	12 (80%)		31 (48%)	20 (63%)		58 (61%)	7 (78%)	
yes	56	44 (54%)	3 (20%)		34 (52%)	12 (37%)		37 (39%)	2 (22%)	
n.a.	12									
Ki67				0.74			0.62			0.03
<20%	109	70 (80%)	12 (75%)		55 (81%)	25 (71%)		83 (81%)	4 (44%)	
≥20%	31	18 (20%)	4 (25%)		13 (19%)	10 (29%)		20 (19%)	5 (56%)	
n.a.	4									
Her2 status				1.00			0.61			0.46
neg	132	85 (97%)	16 (100%)		65 (97%)	34 (94%)		96 (94%)	8 (89%)	
pos	7	3 (3%)	0 (0%)		2 (3%)	2 (6%)		6 (6%)	1 (11%)	
n.a.	5									
Subtype				1.00			0.82			0.03
Luminal A	96	62 (71%)	11 (69%)		48 (71%)	23 (68%)		73 (72%)	3 (33%)	
Luminal B	43	25 (29%)	5 (31%)		20 (29%)	11 (32%)		29 (28%)	6 (67%)	
n.a.	5									

**Table 3 cancers-16-03160-t003:** Correlation analysis between CD66b+ tumor-associated neutrophil (TAN) densities and cell densities of other tumor-infiltrating inflammatory cells in different tumor locations.

		CD66b+ TAN Density			
		Central Tumor Stromal	Central Tumor Intraepithelial	Invasive Front Stromal	Invasive Front Intraepithelial
M1-like (CD68+/CD163−)	Correlation coefficient	−0.020	0.032	−0.099	0.024
	*p*	0.836	0.748	0.295	0.808
	n	107	103	115	103
M2-like (CD68+/CD163+)	Correlation coefficient	0.370	0.239	0.414	0.325
	*p*	<0.001	0.015	<0.001	0.001
	n	107	103	115	103
CD4+	Correlation coefficient	0.475	0.106	0.325	0.081
	*p*	<0.001	0.293	<0.001	0.425
	n	108	100	116	98
CD45RO+	Correlation coefficient	0.240	−0.067	0.038	−0.066
	*p*	0.012	0.506	0.682	0.519
	n	108	100	116	98
CD1a+	Correlation coefficient	0.005	−0.263	−0.055	0.052
	*p*	0.956	0.012	0.563	0.623
	n	107	91	114	93
CD20+	Correlation coefficient	0.343	0.086	0.119	0.061
	*p*	<0.001	0.417	0.207	0.560
	n	107	91	114	93
Correlation coefficient = Spearman’s ρ					

**Table 4 cancers-16-03160-t004:** Univariate and multivariate analysis of disease-free survival according to neutrophil density and clinical characteristics using Cox’s proportional hazards model.

	Univariate Analysis			Multivariate Analysis		
Variable	Hazard Ratio	95% C.I.	*p*	Hazard Ratio	95% C.I.	*p*
Age (yr) (<50 [n = 28] vs. ≥50 [n = 116])	0.96	0.32–2.89	0.947	---	---	---
Stage (UICC I [n = 132] vs. UICC II [n = 12])	2.03	0.59–6.93	0.26	---	---	---
Tumor size (mm) (<20 [n = 134] vs. ≥20 [n = 10])	1.44	0.33–6.19	0.628	---	---	---
Histological grading (G1 [n = 37] vs. G2-3 [n = 104])	6.75	0.91–50.62	0.062	316,875.2	0–>9999	0.96
Histological typing (non-lobular [n = 122] vs. lobular [n = 22])	0.81	0.24–2.79	0.738	---	---	---
DCIS (no [n = 76] vs. yes [n = 56])	0.68	0.26–1.84	0.452	---	---	---
Ki67 (<20 [n = 109] vs. ≥20 [n = 31])	2.24	0.91–5.50	0.078	0.446	0.112–1.782	0.253
Her2 status (negative [n = 132] vs. positive [n = 7])	0.05	0–269.89	0.486	---	---	---
Luminal (A [n = 96] vs. B [n = 43])	1.94	0.80–4.69	0.142	---	---	---
Hormone therapy (No [n = 14] vs. Yes [n = 130])	2.02	0.27–15.06	0.495	---	---	---
Chemotherapy (No [n = 134] vs. Yes [n = 10])	2.38	0.70–8.17	0.167	---	---	---
Stromal CD66b+ TAN density (low [n = 92] vs. high [n = 16])	5.02	1.90–13.24	0.001	4.829	1.2–19.52	0.027
Intraepithelial CD66b+ TAN density (low [n = 69] vs. high [n = 38])	2.98	1.09–8.09	0.033	2.363	0.578–9.66	0.231
Stromal CD66b+ TAN density in lymph nodes (low [n = 103] vs. high [n = 9])	8.49	2.84–25.4	<0.001	8.54	2.13–34.17	0.002

**Table 5 cancers-16-03160-t005:** Univariate and multivariate analysis of disease-free survival according to neutrophil density, macrophage polarization status, and clinical characteristics using Cox’s proportional hazards model.

	Univariate Analysis			Multivariate Analysis		
Variable	Hazard Ratio	95% C.I.	*p*	Hazard Ratio	95% C.I.	*p*
Age (yr) (<50 [n = 28] vs. ≥50 [n = 116])	0.96	0.32–2.89	0.947	---	---	---
Stage (UICC I [n = 132] vs. UICC II [n = 12])	2.03	0.59–6.93	0.26	---	---	---
Tumor size (mm) (<20 [n = 134] vs. ≥20 [n = 10])	1.44	0.33–6.19	0.628	---	---	---
Histological grading (G1 [n = 37] vs. G2-3 [n = 104])	6.75	0.91–50.62	0.062	238,079.58	0–>9999	0.953
Histological typing (non-lobular [n = 122] vs. lobular [n = 22])	0.81	0.24–2.79	0.738	---	---	---
DCIS (no [n = 76] vs. yes [n = 56])	0.68	0.26–1.84	0.452	---	---	---
Ki67 (<20 [n = 109] vs. ≥20 [n = 31])	2.24	0.91–5.50	0.078	1.23	0.41–3.75	0.712
Her2 status (negative [n = 132] vs. positive [n = 7])	0.05	0–269.89	0.486	---	---	---
Luminal (A [n = 96] vs. B [n = 43])	1.94	0.80–4.69	0.142	---	---	---
Hormone therapy (No [n = 14] vs. Yes [n = 130])	2.02	0.27–15.06	0.495	---	---	---
Chemotherapy (No [n = 134] vs. Yes [n = 10])	2.38	0.70–8.17	0.167	---	---	---
Stromal TAM polarization status (other [n = 78] vs. M2-shifted [n = 44])	3.53	1.39–8.99	0.008	1.67	0.56–4.97	0.36
Intraepithelial TAM polarization status (other [n = 84] vs. M2-shifted [n = 34])	5.25	1.96–14.04	<0.001	3.56	1.07–11.81	0.038
Stromal CD66b+ TAN density (low [n = 92] vs. high [n = 16])	5.02	1.90–13.24	0.001	1.67	0.46–6.08	0.44
Intraepithelial CD66b+ TAN density (low [n = 69] vs. high [n = 38])	2.98	1.09–8.09	0.033	1.80	0.56–5.85	0.326

## Data Availability

To prevent possible conclusions on patient identity or medical history by the public, the raw data supporting the conclusions of this article will be made available by the authors on request.

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
