# Peer review of "Tumor-Associated Neutrophils Are a Negative Prognostic Factor in Early Luminal Breast Cancers Lacking Immunosuppressive Macrophage Recruitment"

_cancers, 2024, doi:10.3390/cancers16183160_

Round 1
Reviewer 1 Report
Comments and Suggestions for Authors
The paper, "Tumor-associated neutrophils are a negative prognostic factor in early luminal breast cancers lacking immunosuppressive macrophage recruitment," aims to correlate tumor-associated neutrophil (TAN) counts with breast cancer prognosis. While the data presented in the paper are decent, several of the claims need to be reconsidered or further clarified.
[Line 26] "Our results ... which is a novel finding." It is unclear what specific finding the authors consider novel. The role of TANs has been extensively studied in pathology, with numerous reports debating this correlation. If the authors indeed present a novel finding that previous research has not concluded as significant, it is crucial to discuss why earlier studies did not observe this significance.
[Line 28] "Furthermore, combined analysis with other immune cells from previous studies revealed that intratumoral TANs were only associated with prognosis in tumors that did not express an unfavorable macrophage polarization profile." This is a well-documented finding. The relationship between tumor-infiltrating lymphocytes (TILs), TANs, and macrophages has been extensively explored in the literature. The authors should acknowledge this body of work appropriately.
[Introduction, Line 83] The authors themselves acknowledge that much research has been conducted on TANs, which contradicts the claim of novelty made in the abstract.
[Line 89] The structure of the introduction raises concerns. The authors should provide a comprehensive review of the literature, acknowledging studies that suggest there is no significant prognostic value of TANs. Additionally, they should explain what new parameters or statistical methodologies are introduced in this work that justify revisiting the topic.
"To our knowledge, this is the first study to evaluate the prognostic relevance of neutrophils exclusively in the early luminal subtype." At this point, the authors shift their claim of novelty to the specific focus on the luminal subtype. A more thorough introduction is needed to explain the significance of the luminal subtype and why the broader field has paid less attention to it.
[Results, Line 160] The authors need to provide a better explanation of the previous analyses conducted on these patients. Have there been any conclusions about TANs from this dataset before? Is this dataset part of a larger study?
[Figure 1, Images B and C] The images are unclear and need to be accompanied by a more detailed explanation to guide the reader on what they should be observing.
[3.3 Correlations with Clinical Characteristics, Line 204] Given that the authors have distinctly separated parameters for tumor infiltration, surrounding tissue, and the invasion front, it would be beneficial to compare these. Specifically, when these predictions conflict in terms of prognosis, which parameter is more reliable? How many statistical scenarios are there to analyze?
Overall, the paper is of decent quality and requires minor revisions, most of which can be addressed through careful rewriting and clarification of key points.
Reviewer 2 Report
Comments and Suggestions for Authors "Tumor-associated neutrophils are a negative prognostic factor in early luminal breast cancers lacking immunosuppressive macrophage recruitment" is an interesting study, showing the importance of neutrophils as prognostic factor. There are, hoiwever some concerns, especially in data visualization: 1. In introduction, several sentences about breast cancer, as the major study object would be nice, something like it is done in PMID: 34249678, PMID: 34359725 or PMID: 34926234 2. Figure 1D is totally strange. Actually, as far as I understand you mean that tumor samples have significant difference, however judging by the picture, my conlusion would be that the only significant difference there is in normal tissue.... Either figure has to be fixed, or more explanation should be in the text. 3. Figure 1A is perfect, however the important data in B and C requires increase of PDF to 175% to be readable. 4. Table 3. Shouldn't it have columns? Comments on the Quality of English Languagejust check once more a bit
